# Colchicine combination therapy increases treatment tolerance in patients with arthritis: A systematic review and meta-analysis

Changwei Zhao[1,2], Xiaogang Hao[2], Wenjun Cai[3], Ling-Feng Zeng[4], Wenhai Zhao[2], Xiangxin Li[2]*

1 Changchun University of Chinese Medicine, Changchun, China, 2 The Affiliated Hospital of Changchun University of Chinese Medicine, Changchun, China, 3 Third Affiliated Clinical Hospital to Changchun University of Chinese Medicine, Changchun, China, 4 The Second Affiliated Hospital of Guangzhou University of Chinese Medicine, Guangzhou, China

* 13596109118@163.com

## Abstract

### Background

Arthritis seriously affects people's quality of life, and there is an urgent clinical need to improve the efficacy of medications as well as to reduce the adverse effects induced by treatment. Combined colchicine therapy is gradually being embraced in clinical care, but the evidence remains insufficient.

### Methods

English databases were searched from the establishment to September 4, 2024. Eleven eligible Randomized controlled trials (RCTs) were included. The quality of the literature was assessed by the risk of bias tool in the Cochrane Handbook. Relative risk (RR) and Cohen's d (SMD) were used for categorical and continuous variables, respectively, at 95% confidence interval (CI), and Stata 17.0 software was used for statistical analysis. Sensitivity analyses were used to verify the stability of the analyzed results, and heterogeneity analyses were used to explore the sources of heterogeneity in the studies. Funnel plots and Egger's test were used to assess publication bias.

### Results

Eleven eligible RCTs were included in this study. Compared with conventional treatment, combined colchicine treatment improved patient's global assessment results (SMD = 1.24, 95% CI [0.01, 2.47], P = 0.05, I2 = 0]), stiffness (SMD = -0.81, 95% CI [-1.43, -0.19], P = 0.01, I2 = 63.91%]) and did not increase adverse effects (RR = 0.79, 95% CI [0.31, 1.27], P = 0.36, I2 = 0.00%). However, combined colchicine treatment did not improve visual analog scores (VAS) (SMD = -0.96, 95% CI [-2.85, 0.93], P = 0.13, I2 = 97.99%]), Western Ontario and McMaster Universities Arthritis Index (WOMAC) pain (SMD = 0.01, 95% CI [-0.24, 0.27], P = 0.91, I2 = 0]), WOMAC function (SMD = -0.01, 95% CI [-0.36, 0.16], P = 0.44, I2 = 0]), Total WOMAC scale (SMD = -0.05, 95% CI [-0.33, 0.22], P = 0.70, I2 = 0]), physician's

**Funding:** The author(s) received no specific funding for this work.

**Competing interests:** The authors declare that they have no competing interests, 作者声明他们没有竞争利益.

**Abbreviations:** AS, Ankylosing Spondylitis; BMPs, bone morphogenetic proteins; CI, confidence interval; CK, creatine kinase; GA, gouty arthritis; GA, gouty arthritis; HOA, hand osteoarthritis; IL-18, interleukin-18; IL-1β, interleukin-1 β; MMPs, matrix metalloproteinases; ModHAD, Modified Clinical Health Assessment Questionnaire; MSU, Monosodium urate; NALP3, Nod-like receptor protein 3; NSAIDs, non-steroidal anti-inflammatory drugs; PRISMA, Preferred Reporting Items for Systematic Reviews and Meta-Analyses; RCTs, Randomized controlled trials; RR, Relative risk; SMD, Cohen's d; TGF-β, transforming growth factor beta; TNF-α, tumor necrosis factor-α; VAS, visual analog scores; WOMAC, Western Ontario and McMaster Universities Arthritis Index.

global assessment (SMD = 0.36, 95% CI [-2.27, 3.00], P = 0.79, I2 = 97.04%]) and Modified Clinical Health Assessment Questionnaire (ModHAD) (SMD = -1.72, 95% CI [-4.90,1.45], P = 0.29, I2 = 99.11%]).

## Conclusion

Compared with colchicine alone, combination therapy improves patients' quality of life without increasing the incidence of adverse events.

## Introduction

Arthritis is an inflammatory disease mainly caused by inflammation, infection, degeneration, trauma or other factors in human joints and surrounding tissues. It is characterized by joint redness, swelling, heat, pain, dysfunction and joint deformity, and in severe cases, it could lead to disability. Its pathogenesis is related to autoimmune reaction, infection, metabolic disorders, trauma, degenerative disease and other factors. Based on the etiology of arthritis could be categorized into osteoarthritis, rheumatoid, ankylosing, reactive, gouty, rheumatic, septic and so on [1]. Pathologic changes in arthritis include mononuclear cell infiltration, inflammation, synovial swelling, synovial membrane formation, joint stiffness, and articular cartilage destruction [2]. The incidence of arthritis could gradually increase with age and is most common in women. The prevalence of arthritis also varies among populations with different etiologies [2, 3]. Arthritis has a high prevalence worldwide, with osteoarthritis affecting more than 500 million people worldwide and more than 260 million people suffering from knee osteoarthritis (KOA) [4], and the prevalence of gouty arthritis (GA) in China may even reach 10.47% [5].

Currently, the treatment of arthritis aims to reduce pain caused by joint inflammation, joint wear and tear, and muscle strain [6]. Existing therapeutic agents include analgesics, steroids and non-steroidal anti-inflammatory drugs (NSAIDs), as well as biologically targeted drugs, all of which can alleviate severe pain and inflammation [7, 8]. However, these medications have substantial adverse effects and cannot be used long-term to improve disease symptoms and progression, for example, the side effects of NSAIDs are severe gastrointestinal reactions with, and immune dysfunction and adverse cardiovascular events could occur with biologically targeted drugs [9–11].

Colchicine is a tricyclic lipid-soluble alkaloid that was first isolated from autumn saffron (Colchicum autumnale L., Colchicaceae) in 1820 [12] and in 2009, the U.S. Food and Drug Administration approved it based on clinical trials defining dosage and efficacy [13]. The oral bioavailability of the drug ranges from 24% to 88% (mean 45%) [14]. Currently, clinical application of colchicine is mostly used for the treatment of GA, and its potential mechanism of action and effect on joint tissue metabolism may play a therapeutic role in osteoarthritis [15] and rheumatoid arthritis. To clarify whether patients with arthritis benefit from these colchicine-based treatments, we conducted the study to provide clinical evidence for the use of colchicine-based treatments in patients with arthritis.

## Methods

### Registration

The systematic review and meta-analysis were conducted in accordance with the Cochrane Systematic Review Manual guidelines and registered on the PROSPERO platform

**Table 1. The baseline inclusion characteristics for inclusion.**

| Study ID | Sample size (T/C) | Age Year(T/C) | Intervention | Comparison | Duration of Treatment | Outcomes |
|---|---|---|---|---|---|---|
| CR Davis 2021 [17] | 32/32 | 66± 8/66 ±7 | Colchicine (0.5mg bid) | Placebo tablets | 12 weeks | ①⑨ |
| SK DAS 2002 [18] | 19/17 | 40–75 | nimesulide + Colchicine (0.5 mg bid) | nimesulide + placebo tablets | 12 weeks | ①④⑤⑥⑦⑨ |
| SK DAS 2002 [19] | 19/20 | 54.4±7.9/51.5 ±8.2 | piroxicam + Colchicine (0.5 mg bid) | piroxicam + placebo tablets | 5 months | ①⑦⑨ |
| P Liu 2019 [20] | 52/53 | 43 ± 9/44 ±9 | sodium bicarbonate + Colchicine(0.5 mg, 3 times daily, for 5 days, later changed to once daily) | sodium bicarbonate + etoricoxib | 10 days | ⑨ |
| YY Leung 2018 [21] | 54/55 | 58.9 ± 8.9 / 58.07 ± 8.5 | CT + Colchicine (0.5 mg bid) | CT+ placebo tablets | 16 weeks | ②③④⑥⑦⑨ |
| S. Aran 2011 [22] | 31/30 | 60.23 (7.81) /60.07 (7.89) | CT+ Colchicine (0.5 mg bid) | CT+ placebo tablets | 3 months | ⑤⑥⑨ |
| A. Amirpour 2016 [23] | 32/30 | 35–75 | CT+Colchicine (0.5 mg bid) | CT+ placebo tablets | 4 months | ①②③⑦⑧⑨ |
| Erden M 2012 [24] | 30/30 | 57.6 ± 7.1/ 55.4 ± 6.2 | paracetamol + colchicine(1.5 mg/day) | paracetamol | 6 months | ②③④⑧ |
| Borstad, G 2004 [25] | 21/22 | 63.5/62.5 | Allopurinol + colchicine (0.6mg bid) | Allopurinol + placebo | 6 months | ⑨ |
| McKendry, R 1993 [26] | 15/15 | 40.7(11.8)/40.7 (11.8) | CT + Colchicine (0.6–1.8mg/day) | CT + placebo | 18weeks | ⑧⑨ |
| Pascart, T 2023 [27] | 49/46 | 88/ 87.5 | CT + Colchicine(1.5mg/day) | CT + prednisone | 24h | ①⑨ |

Outcomes: ① VAS; ②WOMAC pain; ③WOMAC function; ④Total WOMAC scale; ⑤Physician's global assessment; ⑥ Patient's global assessment; ⑦ ModHAQ; ⑧ Stiffness; ⑨Adverse events.

Note: CT: Conventional treatment, T: Treatment group, C: Control group

(CRD42023451297). The study was also reported according to the Preferred Reporting Items for Systematic Reviews and Meta-Analyses (PRISMA) statement [16]. The PRISMA checklist is provided in S1 Table 1 in S1 File.

## Search strategy

PubMed, Cochrane Library, Embase, and ClinicalTrials.gov databases were searched independently by CWZ and XGH from the time of inception till September 4, 2024. To prevent search omissions, additional searches were conducted to target RCTs included in previous systematic reviews. Search terms included Arthritis, Arthritides, Colchicine, etc.; S1 Table 2 in S1 File shows the specific search strategy in PubMed.

## Inclusion criteria

The inclusion criteria were as follows: (1) Patients diagnosed with osteoarthritis, gouty arthritis, rheumatoid arthritis, juvenile idiopathic arthritis, psoriatic arthritis, etc.; (2) Intervention: The treatment group was treated with colchicine in combination therapy, and the control group was treated with other non-colchicine therapy, such as conventional treatment or placebo. Conventional treatment includes topical analgesics, supplements, non-steroidal anti-inflammatory drugs and physiotherapy. (3) Outcomes: Primary outcome indicators: VAS score; secondary outcome indicators: WOMAC pain score, WOMAC function score, total WOMAC scale score, the physician's global assessment score, patient's global assessment

score, ModHAQ score, stiffness, incidence of adverse events, and biochemical markers, such as IL-1. (4) Study design: RCTs in English language had no restrictions on blinding; with no restriction on age or race.

### Exclusion criteria

The exclusion criteria were as follows: case-control and cohort studies, case reports, protocols, reviews, commentaries, clinical experience, guidelines, expert consensus, animal or cellular experiments; replicated studies; and literature for which no specific basic information was available when authors were contacted.

### Data extraction

EndNote software (version X9) was used to screen the literature. Literature that initially met the search strategy was screened and data extracted by WJC and LFZ. Firstly, titles and abstracts were read to exclude unqualified literature, and preliminary qualified RCTs was read in its entirety, and literature was screened in strict accordance with inclusion and exclusion criteria and delivered to the third researcher XXL for independent checking. Secondly, the data extraction was conducted including the name of the first author, year of publication, sample size, age, conventional treatment program, intervention group, control group, intervention duration and intervention frequency; primary outcome indicators: VAS score; secondary outcome indicators: WOMAC pain score, WOMAC function score, total WOMAC scale score, the physician's global assessment score, patient's global assessment score, ModHAQ score, stiffness, incidence of adverse events, and biochemical markers, such as IL(interleukin-1)-1.

### Risk of bias assessment

Two researchers CWZ and LFZ assessed the risk of bias of included studies according to the Cochrane Handbook Risk of Bias Assessment Tool. The tool assessed 6 important sources of bias, including randomization process, deviations from the intended interventions, missing outcome data, outcome measurements, selection of the reported results, and overall bias. Risk of bias was assessed for each of the included studies in these seven areas. Each area was assessed as 'high risk', 'low risk' or ' some concerns' by evaluating the completeness of the study and the correctness of the methodology implemented. The two researchers (CWZ and LFZ) operated independently and checked each other. In case of disagreement on the results of the evaluation, A third researcher XXL will participate in the discussion and make the final decision.

### Statistical analysis

The analysis was performed using Stata 17.0 software. A random-effect model was used in case of significant heterogeneity of the pooled studies; otherwise, the fixed-effect model was employed. Cohen's d and relative risk (RR) were used for continuous and categorical variables, respectively, at 95% confidence interval (CI). Heterogeneity was determined using the Q-statistic and $I^2$, with a p-value of $< 0.1$ for the Q-statistic or $I^2 \geq 50\%$ indicating a significant heterogeneity. The L'Abbe plot was used to test heterogeneity among categorical variables. Publication bias was evaluated using funnel plot and Egger test. Sensitivity analyses were performed for studies with significant heterogeneity, and subgroup analyses were performed based on outcome indicators.

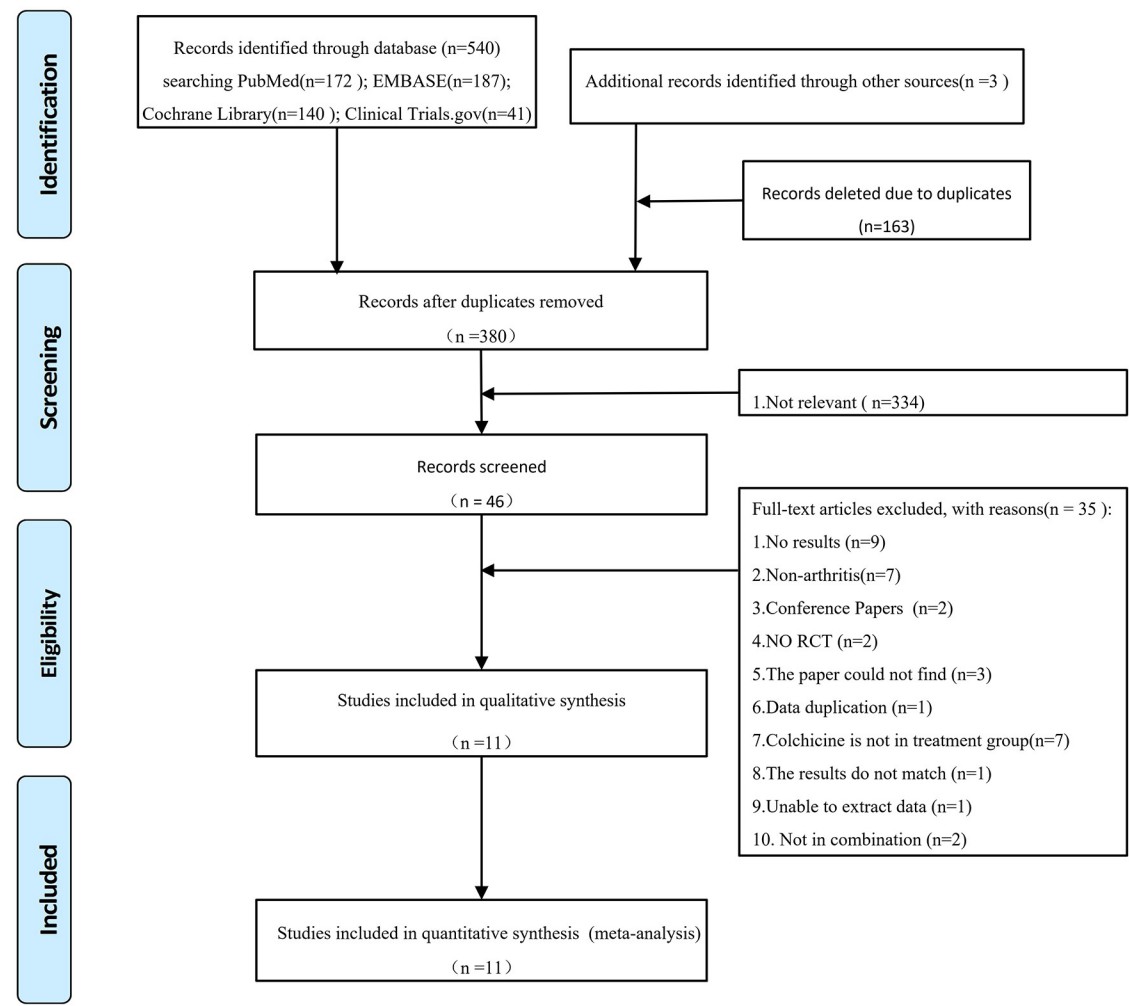

**Fig 1. Flowchart of study selection and identification.**

## Results

### Literature screening

A total of 543 studies were retrieved through the initial search, and after removing duplicates, a total of 380 studies were screened. After screening the titles and abstracts, 46 articles remained; After reading the full text, 11 studies [17–27] were still retained. Fig 1 shows the flow chart of screening.

### Characteristics of the studies

A total of 11 studies [17–27] were included, including one parallel design and one cross-over design experiment, with a total of 704 patients enrolled, 354 in the treatment group and 350 in the control group. Table 1 shows the characteristics of the studies. Some studies had multiple stages or perspectives of indicators analyzed, and the stages with relevant indicators were selected. Among the studies with multiple time periods, two studies [18, 23] data from a shorter period were selected, and. One study [19] included data from week 16. One study [20]

conducted a longer time period. One study [21] performed data from intention to treat. One study [24] got VAS scores based on patient group scores.

## Risk of bias

The risk of bias of 11 RCTs [17–27] was evaluated using the RoB 2.0 tool (Fig 2). 8 studies [17–22, 26, 27] described the method of randomization and was rated at low risk of bias, and the remaining 3 studies [23–25] presented risk concerns regarding the randomization process because of unclear randomization methods. Almost all RCTs showed "low" risk regarding "missing outcome data" except for one study that outcome data were relatively few [25]. There were no cases of selective reporting, deviations from the intended intervention or incomplete results.

## Primary outcome indicators

**VAS score.** A total of 5 studies [17–19, 23, 24] were included, 132 patients in the colchicine group and 129 patients in the non-colchicine group, which showed that there was no significant difference between colchicine and the non-colchicine group in terms of lowering the VAS score (SMD = -0.96, 95% CI [-2.85, 0.93], P = 0.13, $I^2$ = 97.99%]). For the KOA, four studies [18, 19, 22, 24] were included, 100 patients in the colchicine group and 97 patients in the non-colchicine group, and the results showed that there was no significant difference between colchicine and the non-colchicine group in terms of lowering VAS score (SMD = -1.34, 95% CI [-3.63, 0.95], P < 0.001, $I^2$ = 98.05%] (Fig 3). Two studies [26, 27] also mentioned VAS scores, but they could not be pooled, so they were not included in the study.

## Secondary outcome indicators

**WOMAC pain score.** A total of three studies [21, 23, 24] were included, 116 patients in the colchicine group and 115 patients in the non-colchicine group, and the results showed that colchicine had no significant difference in reducing WOMAC pain score [SMD = 0.01, 95% CI (-0.24, 0.27), P = 0.91, $I^2$ = 0] (Fig 4).

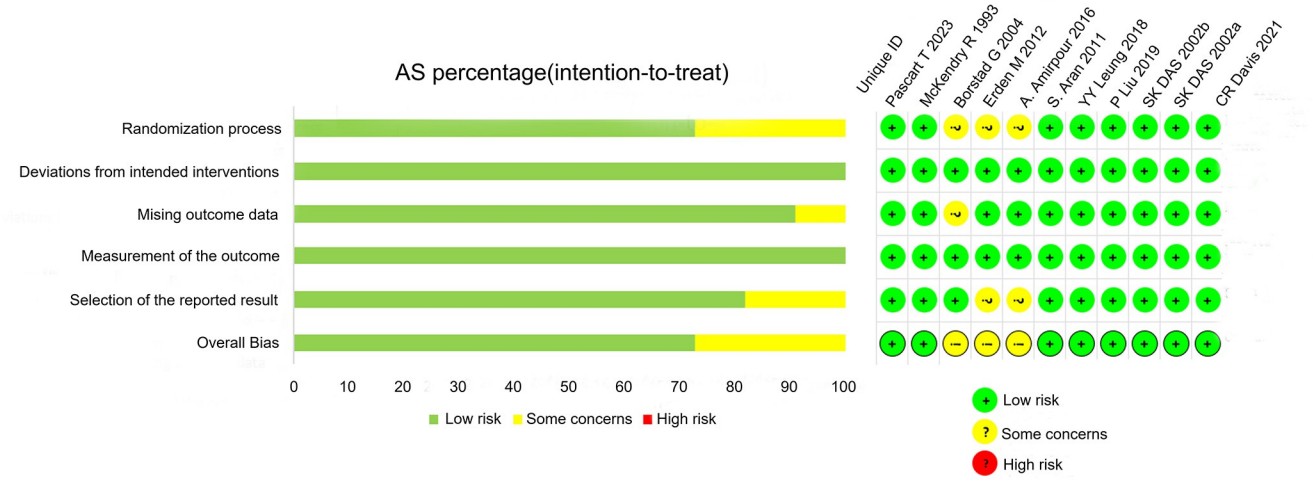

**Fig 2. The risk of bias of 11 RCTs.**

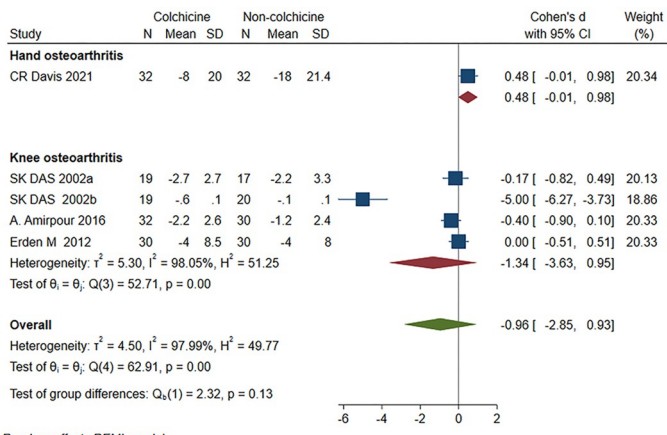

**Fig 3. Comparison of VAS score between the colchicine group and the non-colchicine group.**

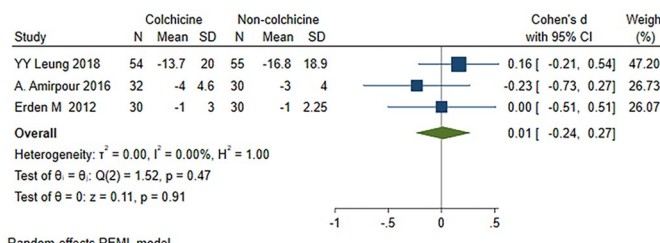

**Fig 4. Comparison of WOMAC pain score between the colchicine group and the non-colchicine group.**

**WOMAC function score.** A total of 3 studies [21, 23, 24] were included, 116 patients in the colchicine group and 115 patients in the non-colchicine group, and the results showed that colchicine had no significant difference in decreasing the WOMAC function score [SMD = -0.10, 95% CI (-0.36, 0.16), P = 0.44, $I^2$ = 0] (Fig 5).

**Total WOMAC scale score.** A total of 3 studies [18, 21, 24] were included, 103 patients in the colchicine group and 102 patients in the non-colchicine group, and the results showed that colchicine had no significant difference in decreasing the total WOMAC scale score (SMD = -0.05, 95% CI [-0.33, 0.22)], P = 0.70, $I^2$ = 0) (Fig 6).

**Physician's global assessment score.** A total of 2 studies [18, 22] were included, 50 patients in the colchicine group and 47 patients in the non-colchicine group, and the results

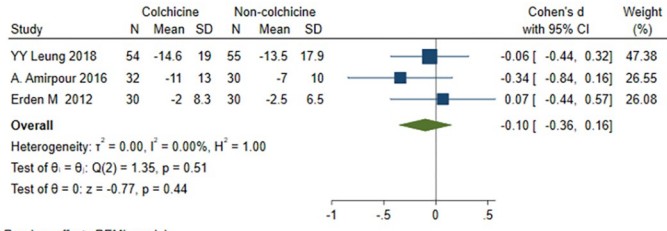

**Fig 5. Comparison of WOMAC function score between the colchicine group and the non-colchicine group.**

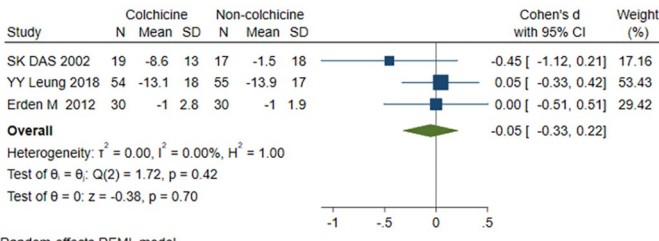

**Fig 6. Comparison of total WOMAC scale score between the colchicine group and the non-colchicine group.**

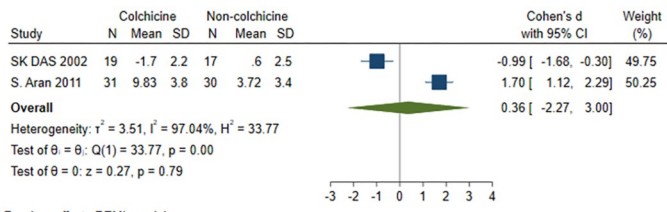

**Fig 7. Comparison of physician's global assessment score between the colchicine group and the non-colchicine group.**

showed that colchicine had no significant difference in decreasing physician's global assessment score (SMD = 0.36, 95% CI [-2.27, 3.00], P = 0.79, $I^2$ = 97.04%) (Fig 7).

**Patient's global assessment score.** A total of 3 studies [18, 21, 22] were included, 104 patients in the colchicine group and 102 patients in the non-colchicine group, and the results showed that the colchicine group was more significant in terms of the patient's global assessment score (SMD = 1.24, 95% CI [0.01, 2.47], P = 0.05, $I^2$ = 92.82%) (Fig 8).

**ModHAQ score.** A total of 4 studies [18, 19, 21, 23] were included, 124 patients in the colchicine group and 122 patients in the non-colchicine group, and the results showed that colchicine had no significant difference in reducing ModHAQ score (SMD = -1.72, 95% CI [-4.90, 1.45], P = 0.29, $I^2$ = 99.11%) (Fig 9).

**Stiffness.** A total of 2 studies [23, 24] were included, 64 patients in the colchicine group and 30 patients in the non-colchicine group, and the results showed that colchicine had no significant difference in reducing stiffness score (SMD = -0.81, 95% CI [-1.43, -0.19], P = 0.01, $I^2$ = 63.91%) (Fig 10).

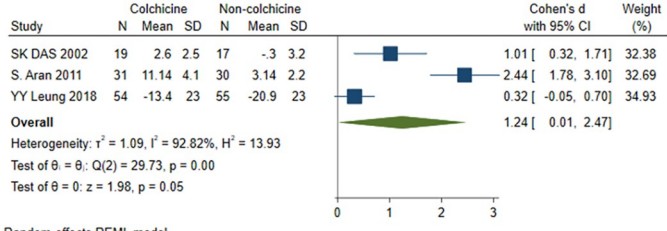

**Fig 8. Comparison of patient's global assessment score between the colchicine group and the non-colchicine group.**

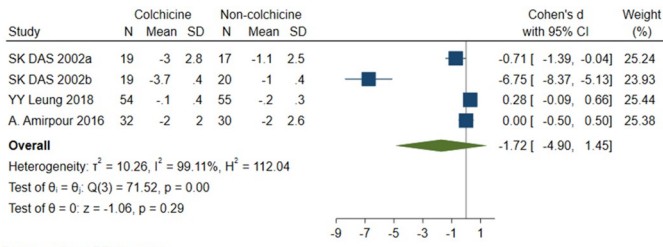

**Fig 9. Comparison of ModHAQ score between the colchicine group and the non-colchicine group.**

**Adverse events.** Adverse events in eight studies [17–23, 25] were included in the calculations, and the results showed that combined colchicine treatment did not significantly increase the occurrence of adverse events (RR = 0.79, 95% CI [0.16, 1.54], P = 0.86, $I^2$ = 0). A total of 5 RCTs [18, 19, 21–23] of osteoarthritis of the knee were included, and the results showed that there was no significant difference between the combined colchicine and non-colchicine treatment groups for adverse events in patients with KOA [RR = 0.85, 95% CI (0.16, 1.54), P = 0.86, $I^2$ = 0] (Fig 11).

## Biomarkers and imaging markers

Ying-Ying Leung et al. reported that colchicine could reduce inflammation and high bone turnover biomarkers known to be associated with knee OA severity and progression risk [21]. In another study in knee OA patients, colchicine lowers whole blood MDA levels but has no effect on whole blood SOD, CAT enzyme activities and GSH levels [24]. In hand OA patients, after 12 weeks of colchicine treatment, CRP and ultrasound synovitis grade showed little improvement [17]. Two studies evaluated the effect of colchicine in GA, and the combination of colchicine treatment showed no significant improvement in uric acid [20, 25], but could upregulate expression of miR-223-3p and miR-451a, downregulated COX-2 and IL-1β [20] and treated with colchicine had fewer acute gout flares [25]. McKendry, R found that psoriatic arthritis laboratory measures were unchanged during colchicine treatment [26].

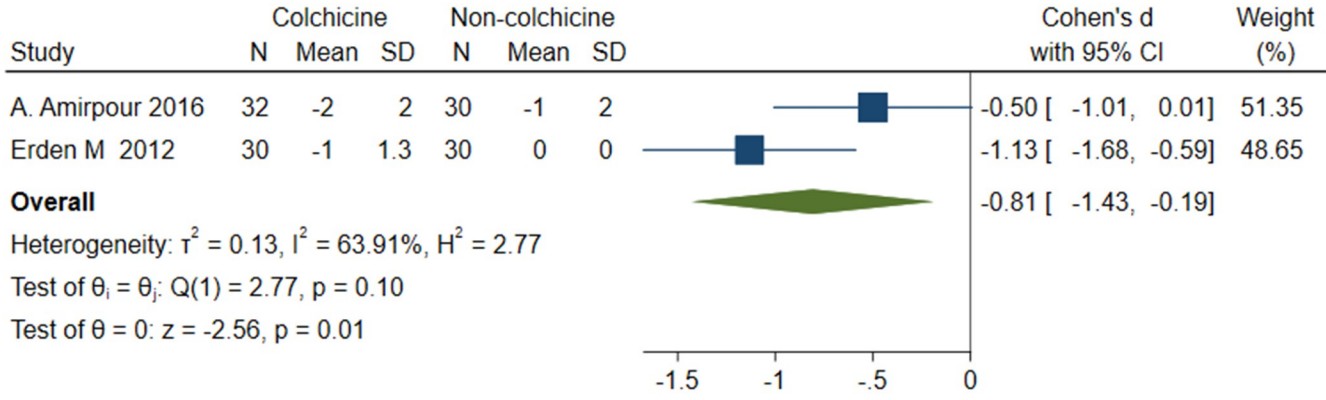

**Fig 10. Comparison of stiffness between the colchicine group and the non-colchicine group.**

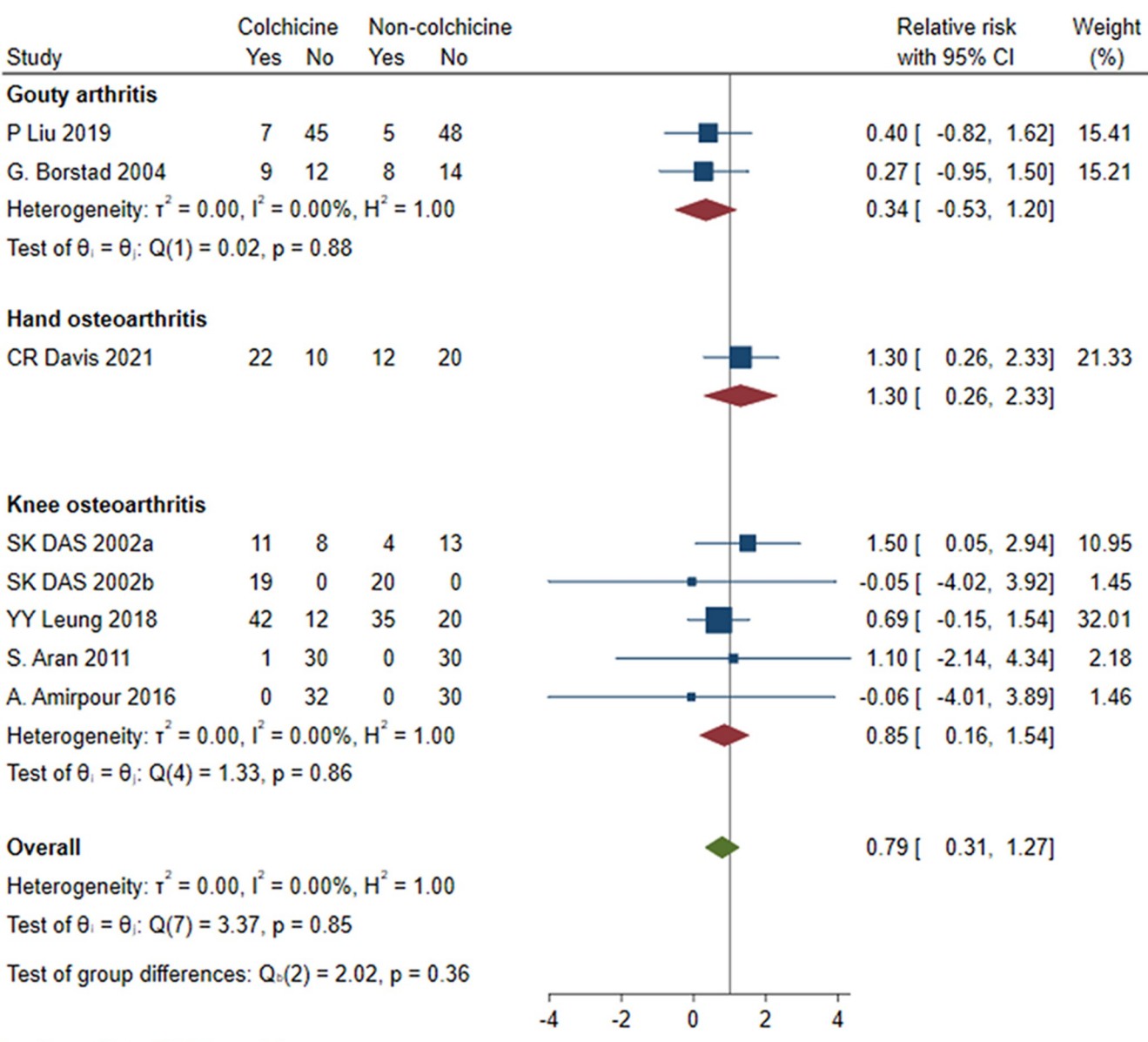

**Fig 11. Comparison of adverse events between the colchicine group and the non-colchicine group.**

## Discussion

Arthritis is a chronic inflammation of one or more joints, with different types of arthritis having different pain characteristics and a long duration of illness. The study screens RCTs on colchicine for the treatment of arthritis, mostly used as an intervention for GA and osteoarthritis. Previous studies have shown that colchicine fails to be beneficial in reducing pain and improving physical function in an overall cohort of patients with the hand osteoarthritis (HOA) or KOA [28], and the colchicine does not increase the incidence of adverse events. However, our study was to evaluate the efficacy and safety of colchicine in the treatment of arthritis and included osteoarthritis, GA, calcium pyrophosphate crystal arthritis and psoriatic arthritis.

This paper suggests that combined colchicine treatment may have an ameliorative effect on patient's global assessment score and morning stiffness and does not significantly increase the incidence of adverse events.

Colchicine is a tricyclic alkaloid that interrupts multiple inflammatory response pathways [13]. Generally speaking, Colchicine interferes with several inflammatory pathways including adhesion and recruitment of neutrophils, superoxide production, inflammasome activation, and nuclear factor κB pathway attenuating the inflammatory response [29]. In our study, colchicine could significantly reduce serum hs-CRP and SF-CTXI in patients with knee osteoarthritis [21].

GA is a disease associated with inflammatory metabolic disorders and is characterized by recurrent episodes of inflammatory arthritis [30]. Patients with GA accompanied by elevated blood uric acid levels may not only be disabled, but may also involve vital organs, leading to complications such as renal impairment, renal failure, hyperlipidemia, hypertension, diabetes mellitus, and atherosclerosis. Uric acid deposits in the joint capsule, bursa, cartilage, bone and other tissues cause damage and inflammation. GA attacks are characterized by a variety of inflammatory cells and cytokines and signaling pathways, such as IL-1β, tumor necrosis factor-α (TNF-α), and other inflammatory mediators [31–33]. Among them, IL-1β is the main mediator that induces GA. Monosodium urate (MSU) crystals deposited in tissues and synovial joints stimulate monocytes to express inflammatory cytokines such as TNF-α, IL-1β, MCP-1, and IL-8, which cause intense inflammatory reactions and result in intolerable joint pain, indicating that urate crystals can promote inflammatory reactions [34]. And neutrophils and related pro-inflammatory cytokines, including TNF-α, IL- 1β and IL-8, play a key role in inflammatory hyperalgesia [35]. However, stimulation of different anti-inflammatory pathways may also play a role in the resolution of gouty attacks. Peroxisome proliferator-activated receptor-γ is activated in gouty attacks, and ligands for this receptor inhibit transcription of genes encoding TNF-α, IL-1, IL-6, cyclooxygenase 2, inducible nitric oxide synthase, and matrix metalloproteinases [36, 37].

Colchicine is a traditional drug for the effective treatment of acute exacerbations of GA [38], and has been widely used in clinical practice. Colchicine binds to microtubule proteins of granulocytes, reduces the adhesion and chemotaxis of neutrophils, inhibits the wandering and aggregation of neutrophils to the inflammatory region, and has a specific effect on GA [38]. By inhibiting Nod-like receptor protein 3 (NALP3) inflammasome and IL-1β expression, it can effectively treat GA, suggesting that the NALP3 inflammasome signaling pathway is one of the important pathways in the inflammatory episodes of GA [39]. In in vitro studies, colchicine could inhibit the activation of MSU crystals in NLRP3 inflammasome, block the release of IL-1β, and inhibit the expression of genes involved in cellular regulation [40]. However, colchicine has a large number of side effects, causing gastrointestinal, hepatic, renal, and central nervous system damage, and even bone marrow suppression with long-term administration [41]. In a study of 185 patients, colchicine was superior to placebo, with no significant difference between the high and low dose groups. There were more associated gastrointestinal side effects in the high dose treatment group [42]. It has also been suggested that traditional high-dose colchicine needs to be used with caution as routine treatment for GA and that low-dose colchicine may be the treatment of choice for acute GA [43]. The European League against Rheumatism 2011 consensus guidelines also recommend the use of low-dose colchicine [44].

KOA is characterized by cartilage degeneration, joint space narrowing, and bone redundancy formation, resulting in joint pain and limited mobility. It is mainly associated with articular cartilage damage, cartilage matrix degradation, subchondral bone remodeling, inflammation and chondrocyte apoptosis metabolism. Inflammatory cytokines [e.g., IL-1β, IL-6, and TNF-α], extra chondral matrix degradation enzymes [e.g., matrix metalloproteinases

(MMPs)], chondrocyte repair and remodeling factors [e.g., bone morphogenetic proteins (BMPs)], and apoptotic factors [e.g., transforming growth factor beta (TGF-β)] are closely involved in articular cartilage damage, subchondral bone remodeling, inflammatory response and apoptotic metabolism [45, 46]. One study showed a strong correlation between synovial fluid uric acid and OA severity confirmed by observation of 159 patients and the possibility that uric acid may be a factor that promotes the pathologic process of OA through activation of inflammatory vesicles and a strong correlation between OA severity and synovial fluid uric acid with synovial fluid interleukin-18 and IL-1β [47]. Some studies have reported that colchicine treatment of patients with osteoarthritis of the knee was significantly better than the corresponding scores of intra-articular steroids and piroxicam alone in the knee pain index (VAS pain) and total KGMC score (modified WOMAC index) of the VAS at weeks 16 and 20, and also showed that the VAS and symptom scores were significantly due to the control group at 20 weeks of colchicine treatment [18, 19, 22]. Colchicine may reduce pain and inflammation in patients with KOA by blocking inflammatory vesicle-mediated pathways. In patients with osteoarthritis without any clinical signs of gout, researchers found that uric acid levels in the synovial fluid were associated with IL-18, IL-1β, and radiographic severity of OA. It is believed to contribute to the local nucleation of MSU crystals. In addition, MSU deposition leading to OA progression may favor OA cartilage degradation. Chondrocyte viability and function are also impaired by MSU deposition, leading to further cartilage damage. This view of OA pathophysiology may confirm the use of colchicine in KOA [21, 48].

As shown in a previous meta-analysis, which analyzed 5 RCTs, there was no statistically significant difference between colchicine and placebo in terms of pain management and functional improvement, even though colchicine is a safe alternative to osteoarthritis of the knee [49]. Another study showed that colchicine failed to achieve the primary outcome at 16 weeks and lacked effectiveness in reducing symptoms and inflammation in KOA compared to placebo [21]. Similarly, we included a 23-week crossover trial that showed no significant difference in colchicine compared to placebo in the treatment of psoriatic arthritis, but more patients preferred colchicine treatment than placebo [26], and we speculate that colchicine may provide transient pain relief and improved quality of life. However, an uncontrolled trial confirmed that 1.5 mg colchicine daily is an effective treatment for psoriatic arthritis [50].

There are few studies discussing the long-term efficacy of colchicine in the treatment of arthritis. Still, a retrospective cohort study in Europe shows that long-term colchicine was the most frequent first-line therapy prescribed for chronic calcium pyrophosphate crystal inflammatory arthritis, which is considered efficient in a third to half of cases [51]. There is also evidence that prophylactic oral colchicine reduces the frequency of recurrence of acute calcium pyrophosphate crystal arthritis [52]. But a 2-day RCT results showed that both oral colchicine and prednisone showed rapid pain relief in patients with acute calcium pyrophosphate crystal arthritis, without significant differences in efficacy [27]. Clinically, colchicine is often used as the first-line drugs for acute GA. However, the long-term efficacy and safety of other arthritis still need to be verified by larger and longer-term clinical observations.

In clinical practice, common adverse events to colchicine include gastrointestinal reactions such as nausea, vomiting, abdominal pain, and diarrhea, elevated creatine kinase (CK), fever, and skeletal muscle headache. The risk of toxicity is increased in the presence of renal failure. We hypothesize that it may be related to the following aspects: firstly colchicine is absorbed in the small intestine and undergoes extensive metabolism, 30% of the absorbed colchicine is distributed to the gastrointestinal tract, muscle, heart, spleen, and leukocytes [53], and 20% of the colchicine is excreted in the urine in the prototype form of the drug, whereas the remaining 50% is metabolized mainly by cytochrome P3A4 in the liver [54]. Secondly, colchicine mainly binds to three proteins in vivo, namely micro tubulin, cytochrome P3A4 and P-glycoprotein

[55], which may result in impaired protein assembly, decreased cytosolization, or morphological alterations, leading to a variety of toxic reactions, and ultimately multi-organ dysfunction [56]. A recent article states that the most significant causes of colchicine poisoning are unauthorized access, intentional overdose, and in-appropriate dosing for gout flares [57]. Therefore, it is essential to use colchicine safely and in moderation.

In our study, the combined use of colchicine mass did not increase the incidence of adverse reactions. This may be related to the dosage used, most of the literature we included was 0.5 mg orally twice daily. Related studies have shown that low doses of colchicine reduce the incidence of adverse reactions compared to regular doses of colchicine [58]. Compared to placebo, low-dose colchicine treatment modestly elevated alanine aminotransferase, albumin, and CK levels, but had no effect on changes in renal function [59]. This implies that combined low-dose colchicine therapy does not increase the incidence of adverse events.

In this study, most of them are high-quality articles, 3 studies [23–25] mention randomness but do not describe the random method, these factors indeed put these studies at a higher risk of bias. The risk of bias could potentially impact the variation in relative risk scores. Studies with inadequate randomization and blinding are more susceptible to systematic errors, which can lead to overestimation or underestimation of the treatment effects. This variability can contribute to the observed heterogeneity in the RR scores.

This study still has some limitations. Firstly, only 11 studies were included, which is a small inclusion of literature; Secondly, the RCTs included were small sample studies; the western drugs used in the control group were not standardized. Thirdly, due to the limited literature included, colchicine for GA and HOA could not be analyzed separately; Fourth, only few types of arthritis are included, others such as rheumatoid arthritis, and juvenile idiopathic arthritis were not included. Finally, there was only one article on calcium pyrophosphate crystal arthritis and psoriatic arthritis, and the duration of treatment for calcium pyrophosphate crystal arthritis was only 24 hours, so we did not combine them. Therefore, we need a larger sample size for further study.

## Conclusion

In combination with colchicine, the quality of life of patients was improved without increasing the incidence of adverse events. More high-quality studies are still needed to provide more reliable evidence for clinical application.

## Supporting information

**S1 File.**
(DOCX)

## Author Contributions

**Conceptualization:** Changwei Zhao, Wenhai Zhao.

**Data curation:** Changwei Zhao, Xiaogang Hao, Wenjun Cai, Ling-Feng Zeng, Xiangxin Li.

**Formal analysis:** Changwei Zhao, Xiangxin Li.

**Supervision:** Wenhai Zhao.

**Writing – original draft:** Changwei Zhao.

**Writing – review & editing:** Changwei Zhao, Xiangxin Li.

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
