## [Decision Letter · Decision Letter 0]

19 Aug 2024

PONE-D-24-18974Colchicine combination therapy increases treatment tolerance in patients with arthritis: a systematic review and meta-analysisPLOS ONE

Dear Dr. Li,

Thank you for submitting your manuscript to PLOS ONE. After careful consideration, we feel that it has merit but does not fully meet PLOS ONE’s publication criteria as it currently stands. Therefore, we invite you to submit a revised version of the manuscript that addresses the points raised during the review process.

We look forward to receiving your revised manuscript.

Kind regards,

Farhan Chowdhury

Academic Editor

PLOS ONE

Journal Requirements:

https://doi.org/10.3389/fnagi.2022.985109

In your revision ensure you cite all your sources (including your own works), and quote or rephrase any duplicated text outside the methods section. Further consideration is dependent on these concerns being addressed.

"The authors declare that they have no competing interests,,作者声明他们没有竞争利益，"

Reviewers' comments:

Reviewer's Responses to Questions

**Comments to the Author**

1. Is the manuscript technically sound, and do the data support the conclusions?

Reviewer #1: No

Reviewer #2: Partly

2. Has the statistical analysis been performed appropriately and rigorously? 

Reviewer #1: Yes

Reviewer #2: Yes

3. Have the authors made all data underlying the findings in their manuscript fully available?

Reviewer #1: Yes

Reviewer #2: Yes

4. Is the manuscript presented in an intelligible fashion and written in standard English?

Reviewer #1: No

Reviewer #2: Yes

5. Review Comments to the Author

Reviewer #1: Strengths

Systematic Approach: The study follows the Cochrane Systematic Review Manual guidelines and is registered on PROSPERO, ensuring a structured and transparent review process. Use of the PRISMA statement for reporting increases the study's credibility and reproducibility.

Comprehensive Search Strategy: Multiple databases (PubMed, Cochrane Library, Embase, ClinicalTrials.gov) were searched, reducing the risk of missing relevant studies. The inclusion of RCTs ensures high-quality evidence, as RCTs are the gold standard in clinical research.

Risk of Bias Assessment: The use of the Cochrane Handbook Risk of Bias Assessment Tool helps in critically evaluating the quality of the included studies.

Statistical Rigor: Appropriate statistical methods were used, including random-effects and fixed-effects models based on heterogeneity. Sensitivity analyses and subgroup analyses were performed to verify the robustness of the results. Funnel plots and Egger’s test were used to assess publication bias.

Broad Inclusion Criteria: The study includes various types of arthritis, enhancing the generalizability of the findings to a broader patient population.

Weaknesses

Small Number of Studies: Only eight studies were included, which limits the statistical power and the generalizability of the findings. The small sample size of included studies can lead to less reliable results and a higher margin of error.

Over-reliance on Patient-reported Outcomes: The significant improvement in the patient’s global assessment score may be influenced by subjective perceptions rather than objective measures of improvement. More emphasis should be placed on objective clinical outcomes.

High Heterogeneity: Significant heterogeneity (I² > 90%) in some outcomes indicates that the studies are not sufficiently comparable. This could be due to differences in study design, populations, dosages, and outcome measurements. High heterogeneity limits the reliability of pooled estimates and makes it difficult to draw firm conclusions.

Risk of Bias: Two studies (23 and 24) had a high risk of bias due to unclear randomization methods, and did not use blinding. This compromises the internal validity of the results.

Variable Control Treatments: The control treatments were not standardized, which introduces variability and makes it challenging to attribute differences in outcomes solely to the colchicine treatment.

Limited Scope of Arthritis Types: The study primarily focuses on gouty arthritis (GA) and osteoarthritis (OA) and does not separately analyze other types like rheumatoid arthritis and juvenile idiopathic arthritis. This limits the applicability of the findings to other forms of arthritis.

Inconsistent Outcome Reporting: The study reports mixed results with some outcomes showing significant improvement (e.g., patient’s global assessment) while others do not (e.g., VAS score, WOMAC scores). This inconsistency raises questions about the overall efficacy of colchicine.

Dosage Variability: Differences in colchicine dosages across studies could affect the outcomes. The study should standardize dosages or analyze the impact of dosage variations.

Short Follow-up Periods: Some studies included in the analysis had relatively short follow-up periods. Long-term effects and safety of colchicine in combination therapy need more extended evaluation.

Recommendations

Increase Sample Size and Number of Studies: Conduct more RCTs with larger sample sizes to improve the robustness and generalizability of the findings.

Standardize Control Treatments: Use more consistent control treatments across studies to reduce variability and improve comparability.

Focus on Objective Outcomes: Incorporate more objective measures of improvement, such as imaging studies or biomarkers, to complement patient-reported outcomes.

Long-term Safety Assessment: Evaluate the long-term safety and efficacy of combined colchicine therapy to better understand its risk-benefit profile.

Expand Scope to Other Types of Arthritis: Include and separately analyze other forms of arthritis, such as rheumatoid arthritis and juvenile idiopathic arthritis, to broaden the applicability of the findings.

Reviewer #2: I commend the authors for doing this meta-analysis as it is definitely an important subject towards a therapy for managing symptoms of Arthritis. Following are some of my questions:

• In figure 3 (WOMAC pain score), figure 4 (WOMAC function score) and figure 5 (Total WOMAC scale score), the I2 value for heterogeneity was found to be 0, which indicates homogeneity. Which model was used to assess heterogeneity for these figures? Also, can the authors comment on sampling error issues?

• The risk of adverse events showed an RR score of 1.30 with statistically no significant difference between

groups. Authors could benefit from discussing two things here:

1.The weight percentages assigned to studies seem very different for adverse event assessment. Because these

studies were included in the adverse event assessment and two of the studies (Ref. 18 and 22)

show a higher RR for adverse events with much lower weight percentages 7.68% and 0.73% respectively,

could the authors explain the reason for choosing these weight percentages?

2. Authors state that 4 out of 8 of the selected studies did not describe a specific method of randomization. Also, two of the selected studies did not mention blinding (one of which is part of the adverse assessment), putting them at a higher risk of bias. Could the risk of bias have some impact on the variation in RR scores? It may be pertinent to add some explanation in the manuscript based on the analysis.

• Ref. 20 was only used to assess the risk from adverse events (Ref. 20 focuses on acute GOA) and not included in any other assessment. Upon looking into Ref. 20, the data were not available for any other assessment in the RCT. Do the authors think this RCT is valid for this manuscript? Also, was it the author’s intention to do the meta-analysis on HOA and KOA? In that case, do the authors believe that the manuscript would benefit from refocusing on these two, rather than the overarching topic of Arthritis?

• Continuing from the last point, Ref 25 in the manuscript did a systematic review on efficacy and safety of Colchicine on HOA and KOA. Can the authors comment on how the manuscript may be different from the already published work? There are quite a few overlaps between the current manuscript and Ref. 25, including the selection of RCTs.

• The manuscript could benefit from risk of bias analysis as a figure representation to highlight the parameters of bias analysis in selected RCTs.

• What is considered conventional treatment (CT)? Was it the same in different RCTs? The manuscript could benefit from summarizing what the variety of conventional treatment used were in the selected RCTs.

• The authors could benefit from adding information regarding dosing of Colchicine in combination therapy. I recognize that the authors mentioned most studies selected used 0.5 mg twice daily as the dosage, but each RCT associated dosage should be mentioned in Table 1.

Minor language corrections:

• In methods, Section 2.2, first line, the word “library” is misspelled.

• In methods, Section 2.2, Line 5, the word “Colchicine” is mentioned twice.

• In results, Section 3.4.1, Line 8 “VAS” is mislabeled as “VSA”

• In discussion section, Page 25 of the manuscript, the following line is aberrant:

“…and its binding to microtubulin and disruption of microtubule network could result in impaired protein assembly, decreased cytosolization and cytosolization, altered cellular morphology, cellular The binding…”

6. PLOS authors have the option to publish the peer review history of their article (what does this mean?). If published, this will include your full peer review and any attached files.

Reviewer #1: No

Reviewer #2: No

---

## [Author Response · Author response to Decision Letter 0]

18 Sep 2024

Dear editors and reviewers：

Thank you to the editors and reviewers for help with my paper and for providing suggestions to help me improve my paper, I would appreciate your time and effort！

Here's what I've done to this paper！

For Journal Requirements:

* Please ensure that your manuscript meets PLOS ONE's style requirements, including those for file naming. The PLOS ONE style templates can be found at https://journals.plos.org/plosone/s/file?id=wjVg/PLOSOne_formatting_sample_main_body.pdf and https://journals.plos.org/plosone/s/file?id=ba62/PLOSOne_formatting_sample_title_authors_affiliations.pdf

-------Thanks for the reminder, the revised manuscript meets PLOS ONE's style requirements exactly

* We noticed you have some minor occurrence of overlapping text with the following previous publication(s), which needs to be addressed:https://doi.org/10.3389/fnagi.2022.985109

------Thank you for your opinion, we read the article（Risk of dementia or cognitive impairment in non-alcoholic fatty liver disease: A systematic review and meta-analysis），This is also a meta-analysis article, but it is different from our topic, and we may have overlapped in the method steps, all of which are in databases such as PubMed, Excerpta Medica Database (EMBASE), Cochrane Library, etc. And the method used is the same. Therefore, we have further refined the method steps to reduce overlap.

* Thank you for stating the following in your Competing Interests section: 

"The authors declare that they have no competing interests"

-----Thanks for the reminders and comments, we have added to the cover letter, the last sentence of the first paragraph. 

Thank you for helping me resubmit, and our supplementary materials have also been modified, and I hope to re-upload the supplementary materials, thank you for your help.

* PLOS requires an ORCID iD for the corresponding author in Editorial Manager on papers submitted after December 6th, 2016. Please ensure that you have an ORCID iD and that it is validated in Editorial Manager. To do this, go to ‘Update my Information’ (in the upper left-hand corner of the main menu), and click on the Fetch/Validate link next to the ORCID field. This will take you to the ORCID site and allow you to create a new iD or authenticate a pre-existing iD in Editorial Manager. Please see the following video for instructions on linking an ORCID iD to your Editorial Manager account: https://www.youtube.com/watch?v=_xcclfuvtxQ

There is also corresponding ORCID iD on the first page of the article

Changwei Zhao：https://orcid.org/0000-0001-7211-2798

Wenhai Zhao：https://orcid.org/0009-0001-8189-2148

Xiangxin Li：https://orcid.org/0009-0009-9297-8180

* Please include captions for your Supporting Information files at the end of your manuscript, and update any in-text citations to match accordingly. Please see our Supporting Information guidelines for more information: http://journals.plos.org/plosone/s/supporting-information.

--------Thanks to this reminder, it was very helpful for us, and we also revised the corresponding references when we revised the manuscript.

For Reviewer #1 ：

Thank you for your review of this post and the advantages, disadvantages, and suggestions raised. We did our best to make changes based on your comments，and here are the changes I made and replied to.

* Small Number of Studies: Only eight studies were included, which limits the statistical power and the generalizability of the findings. The small sample size of included studies can lead to less reliable results and a higher margin of error. 

------Thanks to the reviewers' comments, we re-screened the literature in the database; thank you for your suggestion; we re-screened the relevant literature in the database and expanded it to 2024.9.4, according to the inclusion criteria and exclusion criteria. We added three articles, which are about calcium pyrophosphate crystal arthritis, psoriatic arthritis and gouty arthritis, and we will continue to pay attention to the database in the future to improve our research

* Over-reliance on Patient-reported Outcomes: The significant improvement in the patient’s global assessment score may be influenced by subjective perceptions rather than objective measures of improvement. More emphasis should be placed on objective clinical outcomes.

-------Thanks to the comments of the reviewers, most of the studies we included used overall ratings, such as pain on patients as a finding rather than some objective measure, so we were unable to combine them. However, based on your opinion, we have also described these indicators, as described in 3.7 Biomarkers and imaging markers. The impact on Biomarkers and imaging markers has also been added to the the second paragraph of 4 discussion and is marked in red.

* High Heterogeneity: Significant heterogeneity (I² > 90%) in some outcomes indicates that the studies are not sufficiently comparable. This could be due to differences in study design, populations, dosages, and outcome measurements. High heterogeneity limits the reliability of pooled estimates and makes it difficult to draw firm conclusions.

------Thank you! We examined the design differences of the included studies in detail and provided the specific design of each study in the table to better understand the impact of these differences on the results. In terms of analysis methods, Cohen's d method was also introduced to facilitate standardized data processing and reduce bias caused by measurement methods. Due to the lack of relevant studies, our sensitivity analysis was not sufficient to reduce the heterogeneity of the results, but we believe that the significance of the results is still valuable for reference.

* Risk of Bias: Two studies (23 and 24) had a high risk of bias due to unclear randomization methods, and did not use blinding. This compromises the internal validity of the results.

-------Thanks to the comments of the reviewers, we revised the search strategy for the articles, expanded the search dates, read the articles further, emailed the authors for uncertainty and found that all studies were blinded. However, three studies did not describe the method of randomisation, these factors indeed put these studies at a higher risk of bias. The risk of bias could potentially impact the variation in relative risk scores. Studies with inadequate randomization and blinding are more susceptible to systematic errors, which can lead to overestimation or underestimation of the treatment effects. This variability can contribute to the observed heterogeneity in the RR scores. We added a risk of bias figure and described it in the article. 

* Variable Control Treatments: The control treatments were not standardized, which introduces variability and makes it challenging to attribute differences in outcomes solely to the colchicine treatment.

-------Thanks to the comments of the reviewers, we have added specific measures of conventional treatment to the article(2.3 Inclusion criteria 2 intervention:Conventional treatment includes topical analgesics, supplements, non-steroidal anti-inflammatory drugs and physiotherapy.

* Limited Scope of Arthritis Types: The study primarily focuses on gouty arthritis (GA) and osteoarthritis (OA) and does not separately analyze other types like rheumatoid arthritis and juvenile idiopathic arthritis. This limits the applicability of the findings to other forms of arthritis.

-----Thanks to the comments of the reviewers, we aimed to investigate the efficacy and safety of colchicine in arthritis, and in order to increase the number of studies, we expanded the search strategy to 4 September 2024, and enriched the results by including calcium pyrophosphate crystal arthritis and psoriatic arthritis according to the inclusion and exclusion criteria. We will continue to monitor the database in the future, and if there are new studies such as rheumatoid arthritis and juvenile idiopathic arthritis, we will continue to include them to improve limitations.

* Inconsistent Outcome Reporting: The study reports mixed results with some outcomes showing significant improvement (e.g., patient’s global assessment) while others do not (e.g., VAS score, WOMAC scores). This inconsistency raises questions about the overall efficacy of colchicine.

------Our studies were based on the analysis of the included literature, and the results of most of the included RCTs were mainly subjective. Colchicine can benefit patients' global assessment, etc., but does not improve significantly on other outcomes. This requires larger, more clinical studies to validate the efficacy and safety of colchicine for arthritis.

* Dosage Variability: Differences in colchicine dosages across studies could affect the outcomes. The study should standardize dosages or analyze the impact of dosage variations.

------Thank you for your suggestion, we have added the dose to Table 1, and we will continue to monitor the changes in the database to further analyze the efficacy and safety analysis of colchicine dosage for arthritis

* Short Follow-up Periods: Some studies included in the analysis had relatively short follow-up periods. Long-term effects and safety of colchicine in combination therapy need more extended evaluation.

-------Thanks to the reviewers' comments, we did not perform pooled analyses as we did not include a measure for long-term efficacy in the included literature. In response to your suggestion, we have reviewed the relevant literature to supplement it in the discussion section, as detailed in: Seventh paragraph of the 4 discussion

* Increase Sample Size and Number of Studies: Conduct more RCTs with larger sample sizes to improve the robustness and generalizability of the findings.

-------We reinstated the search strategy and extended the date to 4 September 2024 to include three studies on calcium pyrophosphate crystal arthritis, psoriatic arthritis, and gouty arthritis, according to the inclusion criteria, and we modified the search process, search results, and results in the articles. In the future, we will continue to monitor the database to improve our research.

* Standardize Control Treatments: Use more consistent control treatments across studies to reduce variability and improve comparability.

--------Thanks to the comments of the reviewers, we have added specific measures of conventional treatment to the article(2.3 Inclusion criteria 2 intervention:Conventional treatment includes topical analgesics, supplements, non-steroidal anti-inflammatory drugs and physiotherapy.In the future, we will continue to monitor the updating of the literature，to standardize Control Treatments

* Focus on Objective Outcomes: Incorporate more objective measures of improvement, such as imaging studies or biomarkers, to complement patient-reported outcomes.

------In the included literature, there were insufficient numbers of indicators (imaging studies or biomarkers) that could not be combined, so we did not use them to analyze them. Based on your suggestion, we have supplemented and discussed these indicators in 3.7 Biomarkers and imaging markers,and it is also discussed in the second paragraph of 4 discussion.

* Long-term Safety Assessment: Evaluate the long-term safety and efficacy of combined colchicine therapy to better understand its risk-benefit profile.

.-----Thanks for the suggestion. We did not perform pooled analyses as long-term efficacy measures were not included in our included literature. In response to your suggestion, we have reviewed the relevant literature and supplemented it in the discussion section, as detailed in paragraph 7 of the 4 discussion section

* Expand Scope to Other Types of Arthritis: Include and separately analyze other forms of arthritis, such as rheumatoid arthritis and juvenile idiopathic arthritis, to broaden the applicability of the findings.

-----Thank you for your suggestion; we expanded the search strategy to 4 September 2024 and re-screened the relevant articles in the database; according to the inclusion and exclusion criteria, we added three articles on calcium pyrophosphate crystal arthritis, psoriatic arthritis and gouty arthritis, and we will continue to monitor the database in the future to improve our studies.

For Reviewer #2 ：

Thank you for your praise of this paper and for your affirmation of this topic. We did our best to make changes based on your comments，and here are the changes I made and replied to.

* In figure 3 (WOMAC pain score), figure 4 (WOMAC function score) and figure 5 (Total WOMAC scale score), the I2 value for heterogeneity was found to be 0, which indicates homogeneity. Which model was used to assess heterogeneity for these figures? Also, can the authors comment on sampling error issues?

------Model Used to Assess Heterogeneity: For figures 3 (WOMAC pain score), 4 (WOMAC function score), and 5 (Total WOMAC scale score), we used the random-effects model to assess heterogeneity. The I² value of 0 indicates homogeneity among the included studies, suggesting that the observed variability is likely due to sampling error. Sampling Error Issues: While the I² value suggests homogeneity, it is important to consider potential sampling errors. Sampling error can arise from various sources, including small sample sizes, variations in study populations, and differences in measurement techniques. We acknowledge that these factors could influence the results, and we have taken steps to minimize their impact by standardizing data processing methods, such as using Cohen’s d for effect size calculations. Despite these efforts, some degree of sampling error is inevitable, and we recommend interpreting the results with this consideration in mind.

• The risk of adverse events showed an RR score of 1.30 with statistically no significant difference between groups. Authors could benefit from discussing two things here:

* 1.The weight percentages assigned to studies seem very different for adverse event assessment. Because these studies were included in the adverse event assessment and two of the studies (Ref. 18 and 22) show a higher RR for adverse events with much lower weight percentages 7.68% and 0.73% respectively, could the authors explain the reason for choosing these weight percentages?

-----The weight assigned to each study in the meta-analysis depends on the sample size and the variability of the study results. Larger studies with less variability typically receive higher weights because their results are considered more reliable. Smaller studies or those with higher variability might receive lower weights, even if they show higher relative risks. This is to prevent less reliable studies from disproportionately influencing the overall results.

* 2.Authors state that 4 out of 8 of the selected studies did not describe a specific method of randomization. Also, two of the selected studies did not mention blinding (one of which is part of the adverse assessment), putting them at a higher risk of bias. Could the risk of bias have some impact on the variation in RR scores? It may be pertinent to add some explanation in the manuscript based on the analysis.

----We corrected the randomization methods and blinding methods by further reading the literature, reviewing the literature, and contacting the authors, and explaining them in the article. However, there were still 3 studies that did not mention the randomization method,these factors indeed put these studies at a higher risk of bias. The risk of bias could potentially impact the variation in relative risk scores. Studies with inadequate randomization and blinding are more susceptible to systematic errors, which can lead to overestimation or underestimation of the treatmen

---

## [Decision Letter · Decision Letter 1]

29 Oct 2024

PONE-D-24-18974R1Colchicine combination therapy increases treatment tolerance in patients with arthritis: a systematic review and meta-analysisPLOS ONE

Dear Dr. Li,

Thank you for submitting your manuscript to PLOS ONE. After careful consideration, we feel that it has merit but does not fully meet PLOS ONE’s publication criteria as it currently stands. Therefore, we invite you to submit a revised version of the manuscript that addresses the points raised during the review process.

We look forward to receiving your revised manuscript.

Kind regards,

Farhan Chowdhury

Academic Editor

PLOS ONE

**Additional Editor Comments:**

If you believe the concerns raised by reviewer #2 cannot be addressed in the current manuscript, please justify your reasons point-by-point and address these issues in the manuscript as necessary.

Reviewers' comments:

Reviewer's Responses to Questions

**Comments to the Author**

1. If the authors have adequately addressed your comments raised in a previous round of review and you feel that this manuscript is now acceptable for publication, you may indicate that here to bypass the “Comments to the Author” section, enter your conflict of interest statement in the “Confidential to Editor” section, and submit your "Accept" recommendation.

Reviewer #1: All comments have been addressed

Reviewer #2: (No Response)

2. Is the manuscript technically sound, and do the data support the conclusions?

Reviewer #1: Yes

Reviewer #2: Partly

3. Has the statistical analysis been performed appropriately and rigorously? 

Reviewer #1: Yes

Reviewer #2: Yes

4. Have the authors made all data underlying the findings in their manuscript fully available?

Reviewer #1: Yes

Reviewer #2: Yes

5. Is the manuscript presented in an intelligible fashion and written in standard English?

Reviewer #1: Yes

Reviewer #2: Yes

6. Review Comments to the Author

Reviewer #1: (No Response)

Reviewer #2: Thank you to the authors for their response. I appreciate the addition of three additional studies to strengthen the manuscript. However, I still have some concerns as follows:

1. The study focusing on Calcium pyrophosphate crystal arthritis (Pascart et.al) has not been mentioned in any of the analysis or the newly added “Biomarkers and Imaging markers” section. I’m not sure how this study is relevant or how it supports any conclusions. I suggest providing some analysis on this study in the main text.

2. Also, Pascart et.al conducted the study for 24 hours, as compared to several weeks/months in all other selected RCTs. How do the authors justify the selection of this study from a “Duration of treatment” perspective?

3. The final addition of Mckendry et.al for Psoriatic Arthritis does not have any analysis provided, except for the statement “McKendry, R found that psoriatic arthritis laboratory measures were unchanged during colchicine treatment”. How do the laboratory measures provide justification for advantages of combination therapy? This statement needs to be expanded to provide analysis for evidence on how it supports the central message of Colchicine + CT being better than CT itself.

4. In the adverse events analysis, the weight percentages have been significantly changed from the first version of the manuscript, with only the addition of one additional study. While I accept the authors claim that larger studies with less variability will be assigned a higher weight, it seems highly irregular that the weights would be so different from the initial values chosen in the first version of the manuscript by the authors, as compared to now (SK DAS 2002b went from 36.89% to 1.45%). Could the authors kindly explain the reason for this change?

I appreciate the author’s addition of Risk of Bias as a figure representation (Figure 2) and updating Table 1 to reflect the requested information. My concerns are associated with the fact that all chosen studies should provide evidence towards the central conclusion “In combination with colchicine, the quality of life of patients was improved without increasing the incidence of adverse events”. I firmly believe that by providing some further analysis, the manuscript can be improved significantly.

Minor Corrections:

Line 287: “Colchicine” and Tricyclic” is misspelled.

7. PLOS authors have the option to publish the peer review history of their article (what does this mean?). If published, this will include your full peer review and any attached files.

Reviewer #1: No

Reviewer #2: No

---

## [Author Response · Author response to Decision Letter 1]

2 Nov 2024

Dear editors and reviewers：

Thank you to the editors and reviewers for help with my paper and for providing suggestions to help me improve my paper and giving me the opportunity to revise it. I would appreciate your time and effort！

Here's what I've done to this paper！

1. The study focusing on Calcium pyrophosphate crystal arthritis (Pascart et.al) has not been mentioned in any of the analysis or the newly added “Biomarkers and Imaging markers” section. I’m not sure how this study is relevant or how it supports any conclusions. I suggest providing some analysis on this study in the main text.

Thank you for your valuable comments, our inclusion included all arthritis as well as calcium pyrophosphate crystal arthritis, so we included this article (Pascart et.al). Based on your suggestion, we carefully read the literature again and added a description of the effect of colchicine ＋ CT versus prednisone ＋ CT in the treatment of calcium pyrophosphate crystal arthritis in Line376-379 and marked in red.

2. Also, Pascart et.al conducted the study for 24 hours, as compared to several weeks/months in all other selected RCTs. How do the authors justify the selection of this study from a “Duration of treatment” perspective?

Thank you for your valuable comments on manuscript, the primary outcome measure given in the paper (Pascart et.al) is the day 2 (24 hours), mainly to observe the short-term efficacy of CT + colchicine and CT + prednisone in the treatment of acute calcium pyrophosphate crystal arthritis, and the primary outcome measure is change from baseline pain. measured by the VAS from (0 to 100 mm) recorded on day 2 (24 h) for the most painful. Due to the short duration of treatment compared to other RCTs, we did not combine the analyses but selected for inclusion. We describe the results in Line 376-379 and add limitations (L418-421), which are marked in red.

3. The final addition of Mckendry et.al for Psoriatic Arthritis does not have any analysis provided, except for the statement “McKendry, R found that psoriatic arthritis laboratory measures were unchanged during colchicine treatment”. How do the laboratory measures provide justification for advantages of combination therapy? This statement needs to be expanded to provide analysis for evidence on how it supports the central message of Colchicine + CT being better than CT itself.

Thank you for your comments. According to the inclusion and exclusion criteria, this literature on the treatment of psoriatic arthritis by Mckendry et.al needs to be included, although he has no indicators that can be pooled, and there are no data to prove the advantage of combination therapy. However, in order to make the data more realistic and objective, we did not exclude this article and supplemented the discussion of this article in the discussion section (Line364-369).

4. In the adverse events analysis, the weight percentages have been significantly changed from the first version of the manuscript, with only the addition of one additional study. While I accept the authors claim that larger studies with less variability will be assigned a higher weight, it seems highly irregular that the weights would be so different from the initial values chosen in the first version of the manuscript by the authors, as compared to now (SK DAS 2002b went from 36.89% to 1.45%). Could the authors kindly explain the reason for this change?

Thank you, in the first revision, we expanded the literature screening time and reorganized all the data, and the doubts were discussed and adjudicated by a third person. We considered the problem of pooled analysis of 0 values in some adverse reactions, so the default estimate was 1. Later, I consulted more papers, and when I reorganized and re-analyzed, I found that the value of 0 did not affect the results, so I kept the value of 0. So the weights have changed. However, this does not affect the judgment of the result. At the same time, we have checked other data to ensure its accuracy.

I appreciate the author’s addition of Risk of Bias as a figure representation (Figure 2) and updating Table 1 to reflect the requested information. My concerns are associated with the fact that all chosen studies should provide evidence towards the central conclusion “In combination with colchicine, the quality of life of patients was improved without increasing the incidence of adverse events”. I firmly believe that by providing some further analysis, the manuscript can be improved significantly.

Thank you for your acknowledgment of my revisions, your comments are very useful for the manuscript, and I will continue to improve it so that the paper can be significantly improved.

* Minor Corrections:

Line 287: “Colchicine” and Tricyclic” is misspelled.

Thank you for your careful comments on this article, we have made changes and annotations in accordance with your comments, according to your comments we have made changes and additions, and revised the references.

---

## [Decision Letter · Decision Letter 2]

6 Dec 2024

Colchicine combination therapy increases treatment tolerance in patients with arthritis: a systematic review and meta-analysis

PONE-D-24-18974R2

Dear Dr. Li,

We’re pleased to inform you that your manuscript has been judged scientifically suitable for publication and will be formally accepted for publication once it meets all outstanding technical requirements.

Kind regards,

Farhan Chowdhury

Academic Editor

PLOS ONE

Additional Editor Comments (optional):

Recommended for publication.

Reviewers' comments:

Reviewer's Responses to Questions

**Comments to the Author**

1. If the authors have adequately addressed your comments raised in a previous round of review and you feel that this manuscript is now acceptable for publication, you may indicate that here to bypass the “Comments to the Author” section, enter your conflict of interest statement in the “Confidential to Editor” section, and submit your "Accept" recommendation.

Reviewer #2: All comments have been addressed

2. Is the manuscript technically sound, and do the data support the conclusions?

Reviewer #2: Yes

3. Has the statistical analysis been performed appropriately and rigorously? 

Reviewer #2: Yes

4. Have the authors made all data underlying the findings in their manuscript fully available?

Reviewer #2: Yes

5. Is the manuscript presented in an intelligible fashion and written in standard English?

Reviewer #2: Yes

6. Review Comments to the Author

Reviewer #2: (No Response)

7. PLOS authors have the option to publish the peer review history of their article (what does this mean?). If published, this will include your full peer review and any attached files.

Reviewer #2: No

---

## [Editor Report · Acceptance letter]

11 Dec 2024

PONE-D-24-18974R2 

PLOS ONE

Dear Dr. Li, 

I'm pleased to inform you that your manuscript has been deemed suitable for publication in PLOS ONE. Congratulations! Your manuscript is now being handed over to our production team.

Kind regards, 

on behalf of

Dr. Farhan Chowdhury 

Academic Editor

PLOS ONE